# "Must you make an app?" A qualitative exploration of socio-technical challenges and opportunities for designing digital maternal and child health solutions in Soweto, South Africa

**Sonja Klingberg** [1]*, **Molebogeng Motlhatlhedi**[1], **Gugulethu Mabena**[1], **Tebogo Mooki**[1], **Nervo Verdezoto**[2], **Melissa Densmore**[3], **Shane A. Norris**[1,4], **on behalf of the CoMaCH network**[¶]

**1** SAMRC/Wits Developmental Pathways for Health Research Unit (DPHRU), Faculty of Health Sciences, University of the Witwatersrand, Johannesburg, South Africa, **2** School of Computer Science and Informatics, Cardiff University, Cardiff, United Kingdom, **3** Department of Computer Science, University of Cape Town, Cape Town, South Africa, **4** Global Health Research Institute, School of Human Development and Health, University of Southampton, Southampton, United Kingdom

¶ Membership of the CoMaCH network is mentioned in the Acknowledgments.
* sonja.klingberg@wits.ac.za

**Data Availability Statement:** All data supporting our work are provided within the manuscript.

## Abstract

Participatory and digital health approaches have the potential to create solutions to health issues and related inequalities. A project called Co-Designing Community-based ICTs Interventions for Maternal and Child Health in South Africa (CoMaCH) is exploring such solutions in four different sites across South Africa. The present study captures initial qualitative research that was carried out in one of the urban research sites in Soweto. The aim was two-fold: 1) to develop a situation analysis of existing services and the practices and preferences of intended end-users, and 2) to explore barriers and facilitators to utilising digital health for community-based solutions to maternal and child health from multiple perspectives. Semi-structured interviews were conducted with 28 participants, including mothers, other caregivers and community health workers. Four themes were developed using a framework method approach to thematic analysis: coping as a parent is a priority; existing services and initiatives lack consistency, coverage and effective communication; the promise of technology is limited by cost, accessibility and crime; and, information is key but difficult to navigate. Solutions proposed by participants included various digital-based and non-digital channels for accessing reliable health information or education; community engagement events and social support; and, community organisations and initiatives such as saving schemes or community gardens. This initial qualitative study informs later co-design phases, and raises ethical and practical questions about participatory intervention development, including the flexibility of researcher-driven endeavours to accommodate community views, and the limits of digital health solutions vis-à-vis material needs and structural barriers to health and wellbeing.

**Funding:** This study was funded by the UKRI GCRF Digital Innovation for Development in Africa awards (EP/T030429/1 to MD and NV), Kone Foundation (201901846 and 202105895 to SK), DSI-NRF Centre of Excellence in Human Development at the University of the Witwatersrand (SAN), and Cardiff University's Centre for Artificial Intelligence, Robotics and Human-Machine Systems operation, part-funded by the European Regional Development Fund through the Welsh Government (C82092 to NV). The funders had no role in study design, data collection and analysis, decision to publish, or preparation of the manuscript.

**Competing interests:** The authors have declared that no competing interests exist.

## Introduction

The increased use of information and communication technologies (ICTs) for health, also referred to as digital health, is creating opportunities to support maternal and child health. Digital health has the potential to address health inequalities and community health challenges if undertaken in a contextually relevant manner so as not to exacerbate inequities [1–3]. Existing uses in South Africa and other African countries include reminders about antenatal visits, pregnancy registration, child health information, supporting health behaviours through mobile messages, and technologies specifically adopted in response to the coronavirus disease 2019 (COVID-19) pandemic and its impact on health care services [4–9]. Most notably, South Africa introduced a text messaging service called MomConnect in 2014 for the purposes of registering pregnancies and providing health information to pregnant and postpartum women [10, 11].

Participatory approaches have gained popularity in health research, providing opportunities to incorporate contextual insights and preferences of intended beneficiaries throughout the process of designing studies or health interventions [12]. The rationale for doing so includes ethical arguments for democratic involvement of intended beneficiaries, minimising research waste by ensuring the practical relevance of research outputs to stakeholders expected to implement findings, and improving the quality of said outputs through grounding the research process in lived experiences [12, 13]. In South Africa, participatory design and research approaches have been used to, for example, develop interventions for optimising immunisation coverage and service delivery [14], and to support mothers of pre-term infants [15].

A multi-site study, Co-Designing Community-based ICTs Interventions for Maternal and Child Health in South Africa (CoMaCH), was initiated partly due to the unequal nature of digital health development, taking a participatory and community-centred approach to this domain. The project team includes a multidisciplinary, cross-cultural, and cross-geographical consortium of researchers, technology designers, healthcare professionals and community stakeholders who are exploring the potential of ICTs to enhance maternal and child health and wellbeing in South Africa [16, 17].

This article describes the formative qualitative research phase that was undertaken to contextualise and inform the subsequent co-design process [18] in one of the research sites, the urban township setting of Soweto in Johannesburg Metropolitan Municipality. Previous qualitative research on digital health has been limited in settings like Soweto [19, 20]. A better understanding of the socio-technical challenges and opportunities that digital health technologies afford is needed in the context of maternal and child health in Soweto, especially with regards to the COVID-19 pandemic and resulting shifts to digital technologies [9, 21]. The aim was thus to develop a situation analysis of existing services and the practices and preferences of intended end-users, and explore barriers and facilitators to utilising ICTs for community-based solutions to maternal and child health from multiple perspectives in Soweto. An additional aim of the analysis was to generate transferable insights about the ethics and power dynamics of incorporating participant views into planning and design processes.

## Materials and methods

The qualitative formative phase involved consulting intended users regarding the context, preferences and previous experiences of digital health. Three groups in Soweto were identified as relevant and feasible to reach for gaining multiple perspectives of the context: recent mothers taking part in a separate health intervention trial run by the research team's research unit [22]; adjacent caregivers, meaning family members or other individuals with caregiving roles in the mothers' households; and community health workers employed in the aforementioned trial

[23]. Sampling involved purposive considerations of balancing participant groups, and practical considerations of participants' availability and ability to provide relevant insights. The sample size was pragmatically estimated as a number likely to yield sufficient data for fulfilling the study's aims. Recruitment was carried out by TM, who invited participants and scheduled interviews. Most participants already had experience of research activities run by the research unit, either through previous studies or having worked as affiliated community health workers. Ethical approval was obtained from the University of the Witwatersrand (M200872, MED20-08-043) and Cardiff University, and written informed consent for participation and audio-recording was given by all participants. Each participant received ZAR 156 (approximately US $ 10) to ensure that no costs (e.g. transport) were incurred from participation.

Semi-structured interviews were conducted at the research unit using interview guides (S1 and S2 Files) shared with CoMaCH research sites in other South African provinces, but with openness to emergent, locally relevant topics. Semi-structured interviews were selected to elicit rich accounts of specific topics of interest, while maintaining flexibility beyond pre-determined questions. As the research unit is situated on a hospital campus, and has the capacity to carry out research following COVID-19 protocols (e.g. symptom screening), these interviews were conducted in-person in line with guidance and approvals from the University. All research staff and participants wore masks at all times, in accordance with South African COVID-19 regulations at the time. The interviews were carried out in private rooms with open windows and sufficient space for physical distancing. MM, GM and SK conducted interviews over two weeks in May 2021 with discussions and debriefs between interviews. Due to MM's and GM's working relationships with the community health worker participants, those interviews were carried out in English by SK where possible (7 out of 8 interviews), while MM and GM conducted interviews with ten recent mothers and ten adjacent caregivers, respectively, in English, isiZulu and Setswana as relevant. All interviews were audio-recorded, transcribed verbatim and translated into English where relevant by professional transcribers, and checked and corrected by SK, MM and GM. Audio recordings were stored on a password-protected University server. Copies were securely sent for transcription without any participant identifiers. Some transcription and translation corrections were required, and interpretations arising from the analysis were continuously discussed among the team and checked against recordings to ensure these were not based on incorrectly transcribed or translated data.

To accommodate the multidisciplinary nature of the study, and the three distinct participant groups, qualitative analysis was done using the procedures outlined in the Framework Method [24] approach to thematic analysis [25], which accommodates teamwork and comparisons of different participant sub-groups. The analysis comprised seven stages undertaken collaboratively by the research team: transcription, familiarisation with the interviews, coding, developing a working analytical framework, applying the analytical framework, charting data into the framework matrix, and interpreting the data [24]. SK led the analysis using analysis software MAXQDA, with contributions from MM, GM and TM to coding and the analytical framework, and discussions among the author team regarding the framework matrix and interpreting the data. Coding was initiated through *a priori* codes based on the interview guide and specific topics that the team had identified as relevant during the interviews. As coding progressed, the initial framework was expanded and refined inductively with new and rephrased codes. Based on the final code framework, themes were developed to capture salient insights and patterns of meaning corresponding with the study aim, and illustrative quotes were charted into the framework matrix to convey nuances and contrasts between different participant groups. The analysis predominantly drew on manifest data content. Higher level abstractions relying on more latent subtleties were treated with caution due to the multilingual and cross-cultural dimensions of the study and discussed in depth among the authors.

This qualitative study employs a subjectivist inductive approach, meaning that the general direction is from context-specific qualitative data to theory and practice [26]. Nevertheless, existing theory and pre-determined topics of interest are also relevant to utilise deductively, and the analysis is thus best described as a pragmatic and theory-informing data analysis, as opposed to being either fully inductive theory development or fully theory-informed [26]. The qualitative approaches utilised here involve assumptions about subjective, local and specific reality and knowledge, while also considering material dimensions (e.g. infrastructure). The study therefore draws on critical realist ontology and epistemology without suggesting a hierarchy of the ontological layers of real, actual and empirical [27, 28].

Procedures to ensure the quality and rigor of the research were undertaken in line with Tracy's criteria [29] and reported according to the Standards for Reporting Qualitative Research (SRQR) [30], as detailed in the accompanying checklist. Meaningful coherence with the qualitative research paradigm was prioritised over rigid adherence with standardised criteria. For example, we did not pursue inter-rater reliability or member checking, which involve more positivist assumptions about reality and knowledge [31]. Instead, we focused on ensuring the credibility of the analysis through 'critical friend' approaches within the research team [31].

Due to the study's focus on eliciting the views of intended beneficiaries, a relevant interpretive tool is the typology of participation developed by Arnstein through a ladder analogy [32]. The ladder of citizen participation describes different levels from non-participation through degrees of tokenism to the highest degrees of citizen power and shared control over a process. Arnstein's power-centred theorising has been extended and critiqued in health-related applications [33], but the essence of the ladder as an illustration of power imbalances was taken as a starting point for the analysis of community-based priorities, challenges and facilitators for maternal and child health in the context of societal and health inequalities.

## Results

Twenty-eight adult participants were interviewed. There were no refusals to participate, but some interviews were rescheduled due to participants' other commitments. Ten participants were recent mothers from different neighbourhoods in Soweto, ten were adjacent caregivers (e.g. grandmothers, aunts and sisters from the same household), and eight were community health workers. All participants were Black African women, which reflects the demographics of Soweto and the legacy of colonial and apartheid urban planning. The absence of men as participants is due to both the design of the trial recent mothers and community health workers were recruited from, and the available and actual choices of recent mothers in identifying adjacent caregivers. Through the Framework Method, four themes were developed with different nuances in each participant group, as demonstrated with quotes in Table 1, and pooled descriptions of each theme below.

The themes are formulated as specific conclusions from the analysis, comprising: 1) coping as a parent is a priority; 2) existing services and initiatives lack consistency, coverage and effective communication; 3) the promise of technology is limited by cost, accessibility and crime; and 4) information is key but difficult to navigate. In addition to these four themes, the dataset captured participants' ideas and solutions to maternal and child health, and these are described under the sub-heading of 'Suggested solutions for maternal and child health in Soweto'.

### Coping as a parent is a priority

"*I would help the first-time mothers. . . I can teach them that after birth there is post-natal depression, so to avoid that. . . I would tell them that they must love their children in whatever*

**Table 1. Framework matrix of themes by participant group.**

| Group | Theme | | | |
|---|---|---|---|---|
| | **1. Coping as a parent is a priority** | **2. Existing services and initiatives lack consistency, coverage and effective communication** | **3. The promise of technology is limited by cost, accessibility and crime** | **4. Information is key but difficult to navigate** |
| **Recent mothers (RM)** | Young mothers can be overwhelmed by the new responsibilities and health and welfare needs arising from parenthood, as well as the emotional aspects of parenthood. Suggested solutions often involved community assistance organisations or sharing accurate health information through e.g. community meetings and pamphlets.<br><br>"I think that we must have an organisation that will help kids or mothers. . .so that they can read how they must treat kids, what they must do for a child, so that there must be people that can help you buy other things that you cannot buy for your child, that we cannot do for our kids." (RM2)<br><br>"I never received my parents' love so I didn't know how to love [my child]." (RM7) | Primary health care is relied on but not reliable, and staff do not make sure that health information is understood.<br><br>"They are lazy, they don't want to explain, I don't understand that clinic, when you take your child there to get and injection you stand in a line, take a card, get an injection and then then you leave, they don't explain to you." (RM1)<br><br>"Seriously they're very poor because last time when I took him for the 2-year-old shot he didn't get the injection, they said the government hasn't yet delivered the material." (RM3)<br><br>"Our nurses in our clinics mostly don't know how to communicate with us as parents. Not all of us went to school for knowing how to treat a condition." (RM6) | Mobile data is expensive, which means that many only have data for part of the month. Internet-based solutions are not going to reach everyone, and face-to-face interactions are preferred by many. While public Wi-Fi exists in many parts of Soweto, it is not easily accessible, and hanging around in public to access Wi-Fi is not safe or feasible.<br><br>"Participant: Yes, we buy data on our phones<br>Interviewer: And how long does data last you?<br>Participant: A week, I buy month end" (RM10)<br><br>"I cannot go sit in the streets for me to get access [to Wi-Fi]." (RM9)<br><br>"Okay, to help moms, I think workshops are quite helpful, for face-to-face interactions, and sometimes for, like they show you things, like how they are done. . . Must you make an app?" (RM5) | Written information can be difficult to make use of without clear and friendly guidance. Some recent mothers are actively finding information from different sources but there are also concerns about misinformation online or from other people.<br><br>"So I didn't understand that [maternity case] book to be honest. . .They did not explain anything. . . It's because we didn't ask, actually when you go to the clinic when you are pregnant they give you that paper and then you do what you are supposed to and then you go home, we didn't ask." (RM1)<br><br>"I use these groups on Facebook, mommy groups, groups for moms that have children, then we ask, and then we help each other out." (RM5)<br><br>"Some of us are lazy to go [to clinics], trusting that elders will be there to assist with home remedies, only to find that would worsen the issue." (RM9) |
| **Adjacent caregivers (AC)** | Adjacent caregivers consider young women and recent mothers to have a limited understanding of the responsibilities and practicalities of parenthood but many try to support them where possible even if there is also some judgment involved.<br><br>"Eish ya I would say that to help them we need to tell them that. . . yes a child is a blessing but. . .it's tough these days, so I would teach them that you need to look after the ones you have." (AC1)<br><br>"We would be happy if they could. . .teach us how to raise our grandchildren, what we need to do so that we can understand better." (AC4) | Queuing, rude or too busy staff, and shortages of medicines compromise health care access and waste people's time.<br><br>"Eish it's poor. . .and you wake up early to go to the clinic, by 5:30 you're supposed to have someone join the queue for you, otherwise you'll finish at 4pm, and they're always short-staffed; we need to ask someone to wake up early and go and join the queue for the mother and baby." (AC4)<br><br>"Sometimes you're supposed to go for de-worming, and they don't have de-worming, but if they would have told us via SMS it would be much better than to go and queue. You join the queue when you arrive, and when it's your time to go in there's no de-worming, there's no immunisation." (AC8) | It cannot be assumed that everyone has access to or is able to use a smartphone or the internet even if there is great potential in technology-based solutions and many would prefer them.<br><br>"Honestly, when it comes to phones, I'm not very savvy there." (AC7)<br><br>"I think technology would be better. . . [The clinics] haven't reached that stage of sending WhatsApp (Laughter). . . I'm just saying that they haven't yet reached that stage of sending SMSes, oh no." (AC8) | Information, awareness and education are seen to be the solutions to many health-related issues, but the way in which information is provided matters.<br><br>"Education is the best, it surpasses everything. . . Limited information is dangerous and it doesn't move us forward." (AC4)<br><br>"There are houses where they don't even have TVs, there's no phone because not one person in that house works. So, like if. . . okay in this hall there's such and such a thing on this date, if you could announce that to people and they go and they listen to the presentation, that would be better." (AC5) |

(*Continued*)

**Table 1.** (Continued)

| Group | Theme | | | |
|---|---|---|---|---|
| | **1. Coping as a parent is a priority** | **2. Existing services and initiatives lack consistency, coverage and effective communication** | **3. The promise of technology is limited by cost, accessibility and crime** | **4. Information is key but difficult to navigate** |
| **Community health workers (CHW)** | Community health workers see the challenges recent mothers and young women go through and try to support as much as possible with limited formal welfare, social services or community-based initiatives in place.<br><br>"I think young women especially are facing a lot... How do you provide for those kids, how do you make sure they have a good health, they have good nutrition because that obviously affects their health. So, how do you provide for those things? I think that is the challenge that they face." (CHW2)<br><br>"They need this information, they really need this information because, first of all, when these girls get pregnant they are not so happy they are pregnant, it is most likely it is an accident they are pregnant. So, when they are pregnant they are depressed, they are nervous, they are anxious. They are thinking 'I'm barely making it at home how am I going to take care of a child? ´." (CHW5) | Different solutions are needed for different people and situations because no single system works consistently or for everyone. Community health workers frequently engage in problem-solving to ensure they can help and support trial participants as much as possible despite gaps or challenges.<br><br>"I also think the clinics need to play the part and they are actually not... I have a participant that is 6 months [pregnant], she has not started her clinic visits. She has gone to three different clinics, they send her to this one, to this one, 'no not booking, we are renovating.' And this woman is at risk because she once had a miscarriage so she is high risk already." (CHW2)<br><br>"She was still young and she was worried about her exams, how is she going to go about it... I had to call the school, which is not part of it, but I went the extra mile to call the school and find out the information in how we can assist her after giving birth." (CHW3) | Community health workers are sometimes not able to reach people or capture research data efficiently due to phone network issues and lack of data or electricity. Bringing electronic devices into communities has safety implications even if it facilitates their work.<br><br>"We don't take tablets to the field most of the time because we are trying to be careful, because you must remember that one of our staff members was mugged, her tablet was taken away." (CHW7)<br><br>"It is different each day because today you have network, then by tomorrow you don't. There is no reliability in our work." (CHW2) | Communication needs to be tailored to specific needs and situations, and face-to-face meetings and community events are often preferred for reaching particular groups, supporting mothers and ensuring the information is understood. Using pictures and translating information are also important considerations.<br><br>"I can say home visit is better because others, they will say 'I don't have money for WhatsApp or to buy data'. I think home visit is better because you know how to check her baby, if it is eating, or if she is pregnant... Support is better than just over the phone, yes sometimes you can show the support over the phone but others, they prefer you to come. So, home visit is the best." (CHW7)<br><br>"There should be one person that they can go to and talk to... about their problems and then maybe there will be pictures of communicating maybe, just to say if you are experiencing this, or doing this to the child, just by pictures I think that would be well. And the person should speak the very same language they are speaking so it will be clear." (CHW4) |

*situation they are going to come across... You must love your child and be there for your child... so that your child can grow up in an environment that is right."*

(RM4)

The interviews highlighted how the topic of maternal and child health forms only one part of a complex reality experienced by young women in Soweto. Recent mothers are navigating the emotional dimensions of parenthood along with the more practical aspects. In addition, welfare, income and employment featured strongly in the interviews because both emotional and financial coping are at the core of motherhood for the participants of this study. Notably, this includes concerns around mental health, as evidenced by the quote about first-time mothers and post-natal depression.

While there were differences between how different participant groups described parenting and coping, there was a shared emphasis on young women needing considerable and tailored support. However, adjacent caregivers and community health workers tended to express concerns, and even judgement, towards young mothers in Soweto, describing pregnancies as unwanted.

### Existing services and initiatives lack consistency, coverage and effective communication

*"We know of MomConnect but it has problems so we don't know what is going on... It is not connecting, they are not sending messages. It is just kicking people out. I'm not sure if it is a government problem."*

(CHW1)

There was a recognition among participants that there are many resources and services available, but these do not necessarily cover all the needs of participants or work in a reliable way. Primary health care is relied on for most health-related needs but there are several issues with how services work, and delays in accessing care and medication deter people from spending time trying to get help through clinics. Existing technological solutions, such as MomConnect, are seen as helpful but tend to work inconsistently in practice. Out of the participant groups, the community health workers were the most knowledgeable and optimistic about existing services but recognised the issues mothers may have in accessing these. Recent mothers, more than adjacent caregivers, tended to be familiar with many existing services and solutions but had needs beyond what these currently cover, as described in more detail under the theme about coping as a parent.

### The promise of technology is limited by cost, accessibility and crime

There was a general recognition of technology offering many solutions in terms of efficient communication and information, and its use was seen as somewhat inevitable, as explained by one of the community health workers: *"People tend to respond better if they have technology, and they are using technology so much because people can't live without their phones."* (CHW3)

However, key barriers were highlighted across participant groups in terms of the costs of data, access to and ability to use technology, and the risks of devices being stolen or the need to be in potentially dangerous public spaces in order to access public Wi-Fi. Community health workers knew of more existing digital health solutions and were, for example, more experienced than other groups in using specific websites and online forums for finding health information. Some recent mothers were similarly active in finding information online, but others, especially the adjacent caregivers, were not familiar or comfortable with making use of digital health resources.

### Information is key but difficult to navigate

The ideas participants shared for improving maternal and child health centred on access to information and health education. However, there were also many issues with navigating reliable sources of information or health information being provided in an accessible format, including in the participants' own language. Many therefore suggested alternatives to ICTs in terms of mobilising and reaching communities in more traditional ways such as through meetings, events, pamphlets, flyers, radio or TV.

Concerning the reliability of health information, participants flagged both traditional practices and information on the internet as potentially harmful or inaccurate. Young mothers were not always sure they could trust the advice they received from elders or neighbours, whereas older adjacent caregivers suggested that with more support and information, they could be better equipped to help the younger generations when it comes to parenting and health.

While education and awareness were seen as essential, it was also pointed out that the availability of information does not mean it is well understood, let alone acted upon. People's lack of awareness, or even more judgemental notions of ignorance or laziness, were cited across participant groups as reasons why existing services were not made use of, or why there was a need for more health information. This also reflected a degree of cynicism about information leading to change, as expressed by one recent mother: *"What help is it to know something. . . I am like 'why must I have this information when I won't benefit anything, where I won't help anyone with that', it's just better not to know it."* (RM1) However, it is important to note here that while participants tended to describe individuals' role in navigating information or services, their challenges also reflect broader and structural barriers to accessing or utilising services, as described under the second theme.

### Suggested solutions for maternal and child health in Soweto

Apart from either explicit or implicit references to mental health concerns, recent mothers and adjacent caregivers in Soweto found it challenging to identify specific health challenges in their community when asked directly. Community health workers tended to focus on the topics covered in the intervention they are delivering: nutrition, physical activity, mental health, pregnancy and prevention or management of non-communicable diseases. In general, access to health information, welfare, and any other support required to act on health information were discussed across interviews and topics, but these were not necessarily anchored in any specific aspect of health or illness. The suggested solutions to maternal and child health provided by study participants are described below according to broad categories.

**Digital health solutions.** The range of digital health solutions suggested at this initial phase of the research included: electronic booking systems for clinics; reminders of appointments and children's vaccinations; online support for locating the nearest clinic and checking whether specific services or medications are available; more use of technology and visual aids at clinics (e.g. showing information about anthropometry on screens); improving existing services such as MomConnect; online services for government agencies; digital food assistance vouchers; expanding public Wi-Fi; apps, online groups and forums providing health information and answers to specific question; and, free data for health-related purposes. These suggestions have fed into the co-design phases of the CoMaCH project.

**Workshops and word of mouth.** Due to the challenges related to internet access, and the perceived availability of many people due to high levels of unemployment in Soweto, a typical suggestion was health-themed workshops or other events for providing health education and information, including parenting education and support for recent mothers. In response to the long waiting times at clinics, it was also suggested that such health education sessions could take place at clinics while pregnant women or mothers with their children are waiting to be seen by health care providers.

*"Get workshops, maybe in local schools or local parks, put up tents and invite people. . . People would come through and we would tell them that we are giving out knowledge regarding children's health, mother and child health and whatever you would like to know. . . Word of mouth is for those who don't have smartphones and those who don't have internet access."*

(RM6)

**Home visits, face-to-face support and community engagement.** One solution that recent mothers and adjacent caregivers frequently proposed was home visits by community

health workers, and community health workers themselves also emphasised the value of meeting face-to-face or doing home visits, as opposed to digital health solutions. It is important to note that both public sector and research-affiliated community health workers are deployed across Soweto already, but many participants were unfamiliar with this cadre of health workers.

In addition, participants across the different groups emphasised approaches like going from door to door to share information about upcoming health information events, and the potential to reach many people through community engagement or sensitisation events. Community health workers provided advice on getting buy-in from communities through engaging leaders and designing health promotion efforts in a participatory way: *"When you want to do events, you bring [the community] in to brainstorm… You give the job to the community, and that is how things will work."* (CHW7)

**Flyers, pamphlets, posters and media.**  Many participants mentioned the use of printed information, such as flyers, newspapers, pamphlets and posters, and more interactive solutions such as radio, TV and social media. However, few participants had specific ideas for what approaches would be interesting or engaging enough to get people's attention, and some acknowledged that people may easily ignore information shared via these channels as the theme of information being key but difficult to navigate illustrates.

**Community food assistance and gardening.**  Community-based organisations or churches providing basic food assistance were commonly mentioned as the only existing non-governmental health resources in Soweto, especially in the context of the COVID-19 pandemic and increasing food insecurity due to lockdowns and unemployment. Some of the suggested solutions for promoting maternal and child health therefore also revolved around expanding food assistance, and setting up community gardens on vacant plots of land.

*"The women there have started their own gardens… We can get up and go make our own gardens with beetroot, pumpkin and the like, you know."*

(AC9)

**Group saving schemes and community organisations.**  The interviews included some examples of existing group saving schemes, typically referred to as stokvel in South Africa. These were mentioned as a potential solution for maternal and child health challenges in Soweto through providing financial and social support to families. Similarly, some participants suggested setting up their own community-based organisation to help provide families with childcare, food, children's clothes, and other basic needs, or help people set up small businesses and support each other. There were also calls to improve existing community organisations, for example by ensuring that early childhood development centres have qualified staff.

*"We started a stokvel where we contribute money monthly. You decide how much money you can afford to join the stokvel with, maybe it's ZAR 100, and when that money… comes to you, you can attend to your needs."*

(AC3)

## Discussion

This article provides a situation analysis and explores barriers and facilitators to utilising ICTs for community-based solutions to maternal and child health from the perspectives of recent

mothers, adjacent caregivers and community health workers in Soweto, South Africa. The qualitative analysis resulted in four themes: 1) coping as a parent is a priority; 2) existing services and initiatives lack consistency, coverage and effective communication; 3) the promise of technology is limited by cost, accessibility and crime; and 4) information is key but difficult to navigate. The message is clear that internet-based technology has great potential for health promotion, but it is not consistently accessed by everyone, and thus may fall short of the need to address rather than exacerbate inequalities [1]. This is in line with amplification theory characterisations of technology as enforcing and amplifying, rather than challenging or transforming, existing institutional or structural forces [34].

The findings also echo previous qualitative research from Soweto in that health concerns, especially any future or relatively abstract health issues, must be seen against the background of more urgent personal needs, such as welfare and income, and underlying structural factors that health interventions rarely address [22, 35, 36]. Coping as parents was a common concern with multiple dimensions such as financial, emotional and practical coping. Participants therefore called for material and social support as well as better health information and improved primary health care services.

The experiences captured in Soweto reflect some broader shortcomings of existing digital health resources, and the challenges of navigating what some authors refer to as information ecology [37, 38]. These findings complement evaluations of MomConnect [39, 40] suggesting that there is a general interest in utilising the service if the issues users are facing can be addressed. Identified technical issues include problems with registering for or consistently using MomConnect, and challenges related to language or literacy when engaging with written information [41, 42]. Our participants expressed more critical views of MomConnect than specific evaluations of the programme seem to have received, suggesting that the participants did not perceive the research team to be representing MomConnect, and felt more comfortable being critical about its shortcomings.

These findings echo recognised impediments to trust in, and utilisation of, digital health technologies, including cost, unequal access, defective technology, poor information quality and inadequate publicity hindering optimal use of already existing digital health solutions [9, 43]. Apart from new solutions, enhancing the quality and reliability of existing services would therefore be a promising and economical avenue for improving digital health promotion in South Africa. It may also be necessary to tailor features of existing services from a user experience perspective, as despite the many requests for a service that can provide health information, participants did not seem to be familiar with such features of MomConnect. Indeed, a national evaluation of the interactive features of South Africa's MomConnect mobile messaging programme found that only about 8% of registered users engaged with the programme's features that enabled them to request specific health information [44]. The present study cannot fully explain why utilisation of seemingly promising and acceptable features of existing services remains low, but we recommend engaging users in the development and improvement processes.

While this formative phase involved an openness to emerging ideas and solutions, the scope of solutions was already delineated to digital health, and the topic was broadly described as 'maternal and child health', in line with an international funding call. A fully open-ended participatory process for setting the research agenda or intervention focus [45] was not pursued, in part because of pandemic-related delays and challenges. This raises questions about the extent to which intended beneficiaries' views are listened to and incorporated into the design of research or health interventions [46, 47]. If end-users do not have agenda-setting powers, the scope of solutions is going to be limited to what researchers were able to imagine *a priori* [32, 48]. This is particularly important to consider in a context where the relatively low

priority of health issues vis-à-vis socioeconomic concerns is already well known, and where challenges of technology, including power supply and internet access, easily justify questions like the one posed by a recent mother: "Must you make an app?" (RM5).

Another clash of practical and ethical considerations is the mismatch between participant recommendations, established evidence about health interventions, and problematic histories of paternalistic interventions. For example, participants called for 'educating' the community, and addressing health issues through providing information and raising awareness about specific topics. However, it is a typical assumption that giving people information will lead to health-related behaviour change [49, 50]. Based on decades of health psychology, intervention research and implementation science, some recommendations from participants would likely not lead to effective health interventions or solutions. Nevertheless, public awareness is an important component of wider health promotion efforts [49], and participants' concerns about misinformation or harmful advice should be taken seriously. Finding the balance between listening to participant views, and appropriately incorporating those with existing theoretical and empirical research knowledge is an inevitable challenge of participatory research and health interventions [36, 51, 52]. This, again, points to the responsibility of researchers to also welcome participant views in decision-making, and provide accessible feedback about the process (including why something may not work) in order to be accountable to the groups contributing their time, experiences and ideas to the process.

To return to Arnstein's ladder of citizen participation [32], unless participatory research or intervention design processes are situated on the highest rungs, with participating communities enjoying full authority in terms of setting the agenda according to their needs and preferences, the decision-making will inevitably be dominated by the more powerful. Intentional efforts by research teams, who tend to hold more power than participants, are needed to start bridging such power imbalances. Researchers engaging in participatory or co-design processes thus have a responsibility to communicate clearly and transparently about the topic, the scope of feasible solutions or the process itself, and the extent to which people's real concerns can be accommodated within the parameters of the study or intervention design process [48, 53].

In an effort to address problematic power imbalances and avoid paternalistic health interventions, researchers have drawn on Freire's philosophy [54, 55], aiming to foster a level of critical consciousness about oppressive social structures as a precondition for successful health interventions [56, 57]. Indeed, interventions drawing on notions of critical consciousness have had some success in starting to address structural factors, such as norms around health behaviours, through participatory approaches [58]. Critical consciousness thus enables a recognition of not merely the challenges individuals or communities face, but a deeper understanding of relational factors and power dynamics at play in producing and maintaining (health) inequalities [59]. For developing digital health interventions, critical consciousness could entail awareness and activism around the wider societal inequalities that inevitably influence health promotion efforts, as well as exposure to the more technical and design-specific aspects of digital health [46] (e.g. scale, sustainability and impact [60]). As Fig 1 illustrates, the ladder of participation can incorporate these elements as well, indicating the desired direction towards participation on equal terms. This involves intentional power sharing on the part of the researchers, and critical consciousness on the part of communities, resulting in the ladder analogy becoming more like a staircase.

The participants provided many potential solutions, both digital and beyond, demonstrating the utility of relatively open-ended qualitative research in informing intervention design. Another strength of the study was drawing on diverse perspectives, both through the different participant groups and the multidisciplinary research team composition, as it enabled a

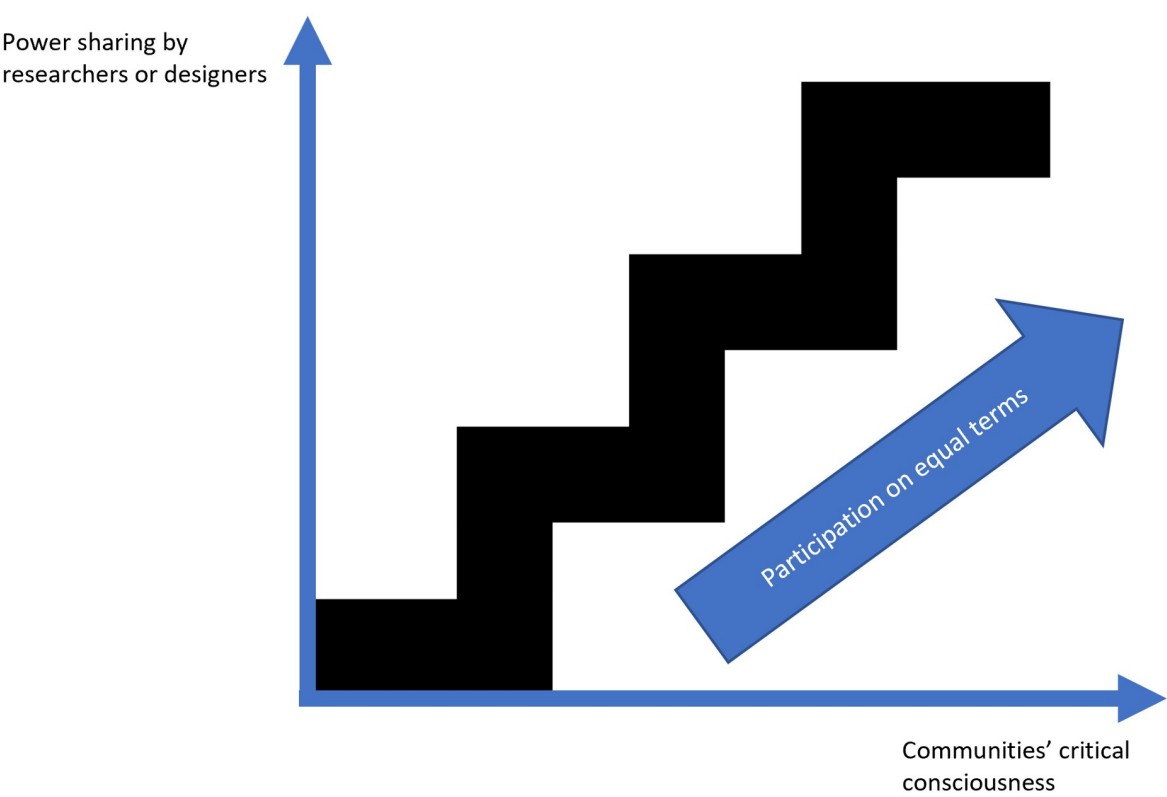

**Fig 1. Staircase of participation [32, 54].**

holistic consideration of what may be relevant, effective or feasible. To honour the participants' contributions, the insights about non-digital solutions are feeding into another Soweto-based research initiative to develop community engagement with health research and interventions [61]. For example, participants' interest in events where reliable health information is provided can be responded to through the research unit's wider community engagement and research dissemination efforts, and community gardening initiatives are being piloted and supported by the unit.

Limitations of the study include data quality concerns due to transcription and translation errors, which are challenging to fully mitigate when several languages are used simultaneously. These challenges have been addressed through continuous discussions about reflexivity and interpretations within the author team [62]. Furthermore, the analysis could have benefitted from more contextual and sociodemographic data. Such data could have provided further nuance but were not collected as the study was not designed for sociodemographic representativeness or sub-group analyses. Another shortcoming is the absence of men among participants, limiting the scope of the study in terms of insights about digital health preferences and utilisation of existing services, as these may involve gendered differences [63]. While men's perspectives were not intentionally excluded, our approach did not involve an active effort to include men, which has been found to be necessary for other qualitative child health research in Soweto [64] and is currently being pursued in other studies at the research unit. The participation and inclusion of men, as well as comparisons between urban and rural research sites, has also been discussed more thoroughly in later phases of the CoMaCH research and co-design process [18].

## Conclusions

This study captured insights about the potential of digital and non-digital solutions for maternal and child health in Soweto. Participatory approaches can help realise such potential in more equitable ways, and this study also sheds light on many context-specific barriers to equitable implementation of digital health solutions. These include factors like the cost of data and limited access to devices, such as smartphones, and underlying socioeconomic circumstances that necessitate more holistic consideration and interventions, such as access to welfare and social support. The findings also highlight the value of conducting formative qualitative research to inform co-design processes for developing health interventions. Further research on men's engagement with digital health in this setting is needed.

## Supporting information

**S1 File. Interview guide for community facilitators.**
(DOCX)

**S2 File. Interview guide for caregivers/guardians.**
(DOCX)

**S3 File. SRQR checklist.**
(DOCX)

**S4 File. Author reflexivity statement.**
(DOCX)

## Acknowledgments

We would like to thank all the participants in the Soweto study and the CoMaCH UK-South Africa Network members: local and international partners and our cross-disciplinary co-investigators. We gratefully acknowledge the feedback from CoMaCH colleagues to the development of this manuscript.

## Author Contributions

**Conceptualization:** Nervo Verdezoto, Melissa Densmore, Shane A. Norris.

**Data curation:** Sonja Klingberg, Molebogeng Motlhatlhedi, Gugulethu Mabena, Tebogo Mooki.

**Formal analysis:** Sonja Klingberg, Molebogeng Motlhatlhedi, Gugulethu Mabena, Tebogo Mooki.

**Funding acquisition:** Nervo Verdezoto, Melissa Densmore, Shane A. Norris.

**Investigation:** Sonja Klingberg, Molebogeng Motlhatlhedi, Gugulethu Mabena, Tebogo Mooki, Nervo Verdezoto, Melissa Densmore.

**Methodology:** Sonja Klingberg, Nervo Verdezoto, Melissa Densmore, Shane A. Norris.

**Project administration:** Molebogeng Motlhatlhedi, Gugulethu Mabena, Tebogo Mooki.

**Resources:** Nervo Verdezoto, Melissa Densmore, Shane A. Norris.

**Supervision:** Sonja Klingberg, Nervo Verdezoto, Melissa Densmore, Shane A. Norris.

**Validation:** Molebogeng Motlhatlhedi, Gugulethu Mabena, Tebogo Mooki.

**Writing – original draft:** Sonja Klingberg.

**Writing – review & editing:** Sonja Klingberg, Molebogeng Motlhatlhedi, Gugulethu Mabena, Tebogo Mooki, Nervo Verdezoto, Melissa Densmore, Shane A. Norris.

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
