## [Decision Letter · Decision Letter 0]

21 Sep 2022

PGPH-D-22-01244

“Must you make an app?” A qualitative exploration of socio-technical challenges and opportunities for designing digital maternal and child health solutions in Soweto, South Africa

Dear Dr. Klingberg,

Thank you for submitting your manuscript to PLOS Global Public Health. After careful consideration, we feel that it has great merit but does not fully meet PLOS Global Public Health’s publication criteria as it currently stands. Therefore, we invite you to submit a revised version of the manuscript that addresses the points raised during the review process. As you will see in the reviews, the reviewers were extremely positive about this work and raised only very minor suggested changes to provide additional context on the respondents.

We look forward to receiving your revised manuscript.

Kind regards,

Hannah Hogan Leslie, PhD

Academic Editor

Journal Requirements:

1.Please provide separate figure files in .tif or .eps format.

Additional Editor Comments (if provided):

Reviewers' comments:

Reviewer's Responses to Questions

**Comments to the Author**

1. Does this manuscript meet PLOS Global Public Health’s publication criteria? Is the manuscript technically sound, and do the data support the conclusions? The manuscript must describe methodologically and ethically rigorous research with conclusions that are appropriately drawn based on the data presented.

Reviewer #1: Yes

Reviewer #2: Yes

2. Has the statistical analysis been performed appropriately and rigorously?

Reviewer #1: N/A

Reviewer #2: Yes

3. Have the authors made all data underlying the findings in their manuscript fully available (please refer to the Data Availability Statement at the start of the manuscript PDF file)?

Reviewer #1: Yes

Reviewer #2: Yes

4. Is the manuscript presented in an intelligible fashion and written in standard English?

Reviewer #1: Yes

Reviewer #2: Yes

5. Review Comments to the Author

Reviewer #1: I recommend that the article be accepted. It is a well written article which addresses current and relevant issues related to the use of digital health to improve access to health information and health care in low and middle income countries. Particularly important is that it explores these issues from the user's perspective.

A clear problem statement and rationale for the study is provided, and the methods are well described. However more could be said about the ethics of doing in-person interviews during the COVID-19 pandemic and precautions taken to protect participants and research staff.

The data analysis and results are well presented, and support the discussion and conclusions.

In reporting and discussing the study findings, it would have been useful to have more sociodemographic information on the participants e.g. education levels, employment. They also appeared to have been participants or community health workers in a related research project. Thus some discussion on how their perspectives may differ from women in other settings such as rural areas in South Africa, or may have been influenced by their proximity to ongoing research on these topic is needed.

A recent systematic review should on the use of digital health technologies in SA during COVID-19 would be good to refer to in the intro and discussion to reflect on whether this study confirms or differs in its findings:

Mbunge E, Batani J, Gaobotse G, Muchemwa B. Virtual healthcare services and digital health technologies deployed during coronavirus disease 2019 (COVID-19) pandemic in South Africa: a systematic review. Global Health Journal. 2022 Mar 5.

Reviewer #2: Congratulations to the authors on this very interesting manuscript that is timely and interesting. The methods, results and discussion flow logically and make sense. The work is important and I am happy to see the in-situ research on the intersection of maternal health and ICT. This is the first review manuscript that I have ever recommended be accepted without modifications.

6. PLOS authors have the option to publish the peer review history of their article (what does this mean?). If published, this will include your full peer review and any attached files.

**Do you want your identity to be public for this peer review?** For information about this choice, including consent withdrawal, please see our Privacy Policy.

Reviewer #1: No

Reviewer #2: No

---

## [Editor Report · Decision Letter 1]

24 Oct 2022

“Must you make an app?” A qualitative exploration of socio-technical challenges and opportunities for designing digital maternal and child health solutions in Soweto, South Africa

PGPH-D-22-01244R1

Dear Dr Klingberg,

We are pleased to inform you that your manuscript '“Must you make an app?” A qualitative exploration of socio-technical challenges and opportunities for designing digital maternal and child health solutions in Soweto, South Africa' has been provisionally accepted for publication in PLOS Global Public Health. Thank you for the prompt and thorough response to the reviewers' comments, and congratulations on an excellent paper - this is certainly the smoothest review process I have had the pleasure of handling!

Best regards,

Hannah Hogan Leslie, PhD

Academic Editor